# Germination: A Powerful Way to Improve the Nutritional, Functional, and Molecular Properties of White- and Red-Colored Sorghum Grains

**DOI:** 10.3390/foods13050662

**Published:** 2024-02-22

**Authors:** Cagla Kayisoglu, Ebrar Altikardes, Nihal Guzel, Secil Uzel

**Affiliations:** 1Scientific Technical Application and Research Center, Hitit University, 19030 Çorum, Türkiye; caglakayisoglu@hitit.edu.tr; 2Department of Food Engineering, Hitit University, 19030 Çorum, Türkiye; ebrar6k@gmail.com (E.A.); nihalguzel@hitit.edu.tr (N.G.)

**Keywords:** germination, sorghum bicolor, FT-IR, SEM, bioactive compounds

## Abstract

This study explored the effects of the germination of red and white sorghum grains (*Sorghum bicolor* [Moench (L.)]) for up to seven days on various properties of the grain. Germination enriched sorghum’s nutritional and sensory qualities while mitigating existing anti-nutritional factors. The study employed Fourier-transformed infrared spectroscopy (FT-IR) and scanning electron microscopy techniques to support its findings. Germination increased protein and lipid content but decreased starch content. White sorghum grains showed elevated calcium and magnesium but decreased iron, potassium, and zinc. Red sorghum grains showed a consistent decrease in mineral content during germination. Germination also increased fiber and lignin values in both sorghum varieties. The results of the FT-IR analysis demonstrate that germination induced significant changes in the molecular structure of white sorghum samples after 24 h, whereas this transformation was observed in red sorghum samples at four days. Total phenolic content (TPC) in red sorghum ranged from 136.64 ± 3.76 mg GAE/100 g to 379.5 ± 6.92 mg GAE/100 g. After 72 h of germination, the germinated seeds showed a threefold increase in TPC when compared to ungerminated seeds. Similarly, the TPC of white sorghum significantly increased (*p* < 0.05) from 52.84 ± 3.31 mg GAE/100 g to 151.76 mg GAE/100 g. Overall, during the 7-day germination period, all parameters showed an increase, and the germination process positively impacted the functional properties that contributed to the health benefits of white and red sorghum samples.

## 1. Introduction

Sorghum (*Sorghum bicolor* [Moench (L.)]) underwent domestication approximately 3000 to 5000 years ago and is currently recognized as the fifth most important crop worldwide in terms of carbohydrate content, following wheat, maize, rice, and barley. Sorghum exhibits remarkable tolerance to drought and serves as a staple food in various semi-arid, arid, and tropical regions globally [1,2,3]. Particularly in dry and semi-arid regions of Africa, Asia, and Latin America, sorghum has widespread use as both animal feed and human sustenance, including bread, snacks, and nonfermented and fermented beverages [1,3,4]. Sorghum is rich in energy, proteins, carbohydrates, vitamins, minerals (such as iron and potassium), and phenolic and flavonoid compounds, which offer potential health-promoting benefits such as anti-carcinogenic, antibacterial, and antioxidant properties, making it suitable for human consumption [3]. The proximate composition of sorghum varies across different varieties. Certain sorghum types, such as red, brown, and black varieties containing color pigments in the pericarp layers, have garnered attention due to their higher phenolic content (generally comprising phenolic acids, flavonoids, and condensed tannins) [5]. Despite its variations, sorghum typically contains significant amounts of macronutrients, including lipids (1.6–6%), proteins (7–15%), and carbohydrates (50–60%). A 100 g serving of sorghum, similar to maize and wheat, provides substantial amounts of resistant starch and can supply approximately 400 calories [3,6]. Furthermore, its lack of gluten proteins makes it a viable alternative for individuals with celiac disease or wheat allergies [3,7,8,9]. 

However, sorghum’s nutritional components, particularly its proteins, are less digestible for humans and monogastric animals due to anti-nutritional factors like tannins and phytic acid. To enhance the nutritional value of sorghum and fully exploit its potential as food or animal feed, it is essential to mitigate these undesired components. Tannins and sorghum proteins interact to decrease protein and starch digestibility. This interaction is significant for human and animal nutrition, as sorghum proteins and tannins can form complexes that render proteins indigestible and inhibit digestive enzymes [8,10]. Tannins, including hydrolyzable (e.g., gallic tannins and ellagitannins), condensed (e.g., proanthocyanidins), and complex tannins, are water-soluble phenolic compounds. Condensed tannins can bind to and precipitate protein components, making them unavailable and indigestible. Moreover, tannins can bind not only to proteins but also to minerals, thus rendering them inaccessible, and they can inhibit hydrolytic enzymes such as trypsin, α-amylase, glucoamylase, and lipase. Another anti-nutritional factor in sorghum is phytate, which forms insoluble complexes with proteins and mineral cations, reducing the bioavailability of trace minerals and protein digestibility [1,3,7]. Therefore, various approaches such as fermentation, germination, and hydrolysis are employed to convert bound phenolic compounds into free forms and enhance the digestibility of protein and starch [11]. 

Germination represents a cost-effective and efficient technology that facilitates structural modifications and the synthesis of new compounds with heightened biological activity, improved nutritional value, and enhanced grain stability. Germination is considered a complex, selective, and efficient method for improving the nutritional and functional value and also reducing the anti-nutritional factors of grains and legumes. During germination, a large number of bound enzymes stored in dry plant seeds are activated, and macromolecular storage substances (starch, proteins, and fat) are hydrolyzed and transformed into more digestible forms. Meanwhile, anti-nutrient components such as tannins and phytic acid can be reduced thanks to hydrolysis triggered by germination. Germination leads to the accumulation of bioactive compounds that are beneficial to human health, such as Ɣ-amino butyric acid (GABA), polyphenols, flavonoids, and vitamins [12].

Germinated sorghum and its flour are suitable for producing specialty foods and value-added products. In comparison to native sorghum, germinated sorghum is considered significantly healthier since the germination process reduces or eliminates anti-nutritional factors such as tannins, phytates, and protease inhibitors, thereby improving nutrient digestibility and vitamin and mineral availability and generating various bioactive compounds that promote health. The germination of sorghum leads to increased levels of albumin, globulin, free amino nitrogen, protease, and kafirin. Additionally, it enhances the riboflavin availability, nutrient digestibility, antioxidant activity, and sensory acceptability of sorghum-based products [3,7]. 

The present study aims to subject red and white sorghum varieties currently limited to animal feed use, especially in our country, Turkey, to the germination process. This endeavor seeks to expand their utilization in human nutrition and investigate the effects of germination at different time intervals on the functional composition of sorghum.

## 2. Materials and Methods

### 2.1. Materials and Reagents

The red and white sorghum (*Sorghum bicolor* L.) grains were provided by a commercial company (Ingro, Turkey). The sorghum grains were cleaned of impurities by sieving and kept at a refrigerator temperature (+4 °C) until the germination process. The germination process was carried out in two replications. All chemicals used in the study were of analytical purity.

### 2.2. Germination Process

Germination was accomplished as determined by [3] with some modifications. Ungerminated red and white sorghum grains were designated as Control Groups and labeled as C_R_ and C_W_, respectively. Approximately 100 g of sorghum samples were rinsed three times with distilled water (1 min each). After that, the grains were soaked in distilled water at a ratio of 1:1 (sample to water) and soaked for 12 h at ambient temperature (23–25 °C). Thanks to soaking, the microbial load was decreased, and the grain moisture level was optimized between 35% and 40%, which is suitable for germination. Following the filtration of the soaking water, the Soaked Control Groups were labeled as SC_R_ and SC_W_ (Soaked Control Red and White Sorghum). Subsequently, the soaked samples underwent a germination process for a duration of 7 days, with sampling conducted every 24 h. During germination, the grains were moistened with water once a day. The samples taken from different intervals of germination were lyophilized. After lyophilization, the germinated and ungerminated (control) samples were milled in an analytical hammer mill (Brabender Laboratory Mill, (Duisburg, Germany) equipped with a sieve aperture of 500 μm. The samples were packed hermetically and finally stored at a refrigerator temperature (+4 °C) until further analysis. The germination process for each sorghum sample was performed in two replications.

### 2.3. Proximate Composition

The moisture content of the samples was determined using a halogen moisture analyzer (Shimadzu MOC63U, Kyoto, Japan) according to the measurement conditions—set as 5 g of the sample at 130 °C. 

Samples (~500 g) were scanned using a near-infrared transmission spectroscopy (NIR) Perten Grain Analyzer (Model DA 7250, Perten Instruments, Springfield, IL, USA) to determine the protein, starch, ash, and lipid contents. Before scanning, the NIR device was calibrated and validated with the results from wet chemical testing of 50 samples of each trait scanned. 

### 2.4. Determination of Mineral Element Composition Using Inductively Coupled Plasma Emission Spectrometry (ICP-OES)

Macroelements (Ca, K and Mg) and microelements (Fe and Zn) were determined by an Inductively Coupled Plasma-Optical Emission Spectrophotometer (ICP-OES, ICAP6500, Thermo Scientific, (Waltham, MA, USA) according to the method described by [13]. The ICP-OES operating conditions are given in Table 1. A closed microwave digestion unit (Berghof Instruments, Speedwave, Germany) equipped with Teflon vessels was used to mineralize 0.30 g of each sample, to which 5.0 mL of ultrapure 65% m/v nitric acid and 2 mL of hydrogen peroxide 35% m/v were added, in order to determine the dissolved metals and non-metals. The sample solution was filtered through a membrane filter (pore size 0.45 µm) and filled to an exact volume of 0.25 mL.

Calibration samples were prepared from multi-element standard solutions (100 mg mL^−1^). Standard solutions (Chem-Lab, Zedelgem, Belgium) were diluted to concentrations in a range between 0.1 and 10 mg/L for microelements and between 1 and 40 mg/L for macroelements. 

### 2.5. Crude Fiber Content

The Acid Detergent Fiber (ADF) and Neutral Detergent Fiber (NDF) composition of the samples were determined according to the modified method of [14]. 

### 2.6. Fourier Transform Infrared (FT-IR) Spectroscopy Characterization

Chemical changes in sorghum samples due to germination were evaluated using a nitrogen-cooled Attenuated Total Reflectance FTIR (Thermo Scientific Nicolet Nexus 470, Waltham, MA, USA) equipped with a mercury/cadmium/telluride detector at a resolution of 4 cm^−1^ [15]. The spectra of the samples were scanned in the range 4000–400 cm^−1^. Data were processed using Omnic software (OMNIC 9.0). 

### 2.7. Scanning Electron Microscopy (SEM)

The morphological properties of the germinated sorghum grains were determined using a scanning electron microscope (ANATEK, FEI/Quanta 450 FEG, CZECHIA). The samples that were coated with a thin film of gold were mounted on an aluminum stub plated with double-sided adhesive tape and examined at 10,000 kV of accelerating voltage. The magnification value, pressure, and scale were set as 1000×, 3.04 × 10^−3^ Pa, and 100 µm, respectively. 

### 2.8. Color Measurement 

The CIE color parameters of soaked, germinated, and control samples were measured using a Minolta CM-3600d (Minolta, Osaka, Japan) colorimeter as described by [16].

### 2.9. Determination of Total Phenolic Content

The total phenolic content (TPC) was measured by a Folin–Ciocalteu assay [17] with minor modification according to [18]. Sorghum extracts were obtained by finely grinding wholegrain sorghum in a laboratory mill (Brabender, Laboratory Mill SM4, Duisburg, Germany) and passing it through a 0.5 mm sieve. Then, 50 mg of grounded grain sample was mixed with 1.5 mL of extraction solvent of methanol, water, and hydrochloric acid solution (50:48.5:1.5, *v*/*v*/*v*). Extraction was carried out in an ultrasonic bath (Sonorex DigiPlus DL 255 H, Bandelin, Taufkirchen, Germany) at a 37 kHz frequency, 25 ± 2 °C, and 100% amplitude for 30 min. The extract was then centrifuged (Sigma 3K30, Taufkirchen, Germany) at 5000× *g* for 15 min. The extraction process was repeated, and the supernatants were combined until the last extraction volume was reached at 5 mL for white sorghum and 10 mL for red sorghum samples. The extraction procedure was performed in duplicate for all samples. A 500 µL of extract was mixed with 500 µL of Folin’s reagent (0.2 N) and 1 mL of Na_2_CO_3_ solution (7.5%). The mixture was incubated at room temperature for 60 min in the dark. Absorbance measurements were taken at 720 nm (Shimadzu UV-1800, Kyoto, Japan), and results were expressed as mg Gallic Acid Equivalents (GAE)/100 g dry weight (R^2^ = 0.9973).

### 2.10. Determination of Radical Scavenging Activity

The 2.2-diphenyl-1-picrylhydrazyl (DPPH) radical-scavenging activity assay was performed according to the method described by [19], with minor modifications. Then, 100 μL of the extract was mixed with the DPPH solution (0.1 mM) of 1900 μL that was allowed to stand for 30 min at room temperature in the dark. The absorbance was measured at 515 nm. The results were expressed as mg Ascorbic Acid Equivalents (AAE)/100 g dry weight (R^2^ = 0.9988).

### 2.11. Statistical Analysis

All measurements (mean ± standard deviation) were performed in triplicate. Statistical analyses were performed using one-way ANOVA (SPSS’16) at a significance level of 0.05. Tukey’s test was used to differentiate between mean values. 

## 3. Results and Discussion

### 3.1. Germination Process and Proximate Composition

The germination process for a duration of 7 days (Figure 1) resulted in significant changes in the chemical composition of sorghum grains. The change in the proximate composition (moisture level, protein, ash, lipid, and total starch contents) of red and white sorghum grains is given in Table 2.

In Table 2, it becomes apparent that white sorghum grains exhibit a more abundant composition regarding ash, protein, and lipid content. The protein content of both samples increased following a 7-day germination period, with a notable rise of 23% in red sorghum and 19% in white sorghum. Previous studies in the literature have indicated that this increase in protein content is attributed to the elevation of specific amino acids such as lysine, valine, phenylalanine, methionine, and tryptophan during germination [4]. Similar findings were reported by [1], who observed a significant increase in protein content after applying the germination process to corn in a study resembling ours. Furthermore, [20] conducted a study on quinoa germination (72 h) and found that protein content increased (control sample: 9.6 g/100 g, 72 h sample: 26 g/100 g), while lipid content decreased (control: 15.2 g/100 g, 72 h germinated sample: 7.6 g/100 g). 

The increased crude protein content during germination can be explained by the production of enzymes by the developing seed, compositional changes resulting from the breakdown of other elements, and the synthesis of newly formed proteins. For instance, α-amylase enzymes break down starch granules, releasing packed proteins from the seed structure. Additionally, increased protease enzyme activity during germination leads to the breakdown of peptide components into amino acids, thereby increasing the protein content of germinated grains. Respiration during the germination process contributes to the biological synthesis of new amino acids, while losses in dry matter, particularly carbohydrates, also play a role in protein content enhancement [20,21].

Regarding other compositional changes, both samples exhibited an increase in ash, moisture, and lipid content during germination, while starch content decreased. The germ of grain seeds contains the majority of crude lipids, and during germination, enzymes hydrolyze triacylglycerol, resulting in the production of free fatty acids. These fatty acids undergo β-oxidation in the cytosol and mitochondria, generating vital energy to support seed development. Consequently, a decrease in crude fat content is expected during germination. However, in our study, an increase in lipid content was observed, which differs from the decrease reported in legume germination [20]. During the germination of sorghum, the increase in fat content can be attributed to various factors. Firstly, the mobilization of stored reserves within the seed, including lipids, proteins, and carbohydrates, provides energy for the growing seedling. As the seed undergoes metabolic processes, the breakdown of stored proteins and carbohydrates may lead to the synthesis and accumulation of fats. Additionally, the activation of lipid biosynthesis pathways during germination contributes to the production of new lipids required for cell division and expansion. Overall, the increase in fat content during sorghum germination reflects complex biochemical processes essential for seedling growth and development.

Moisture content decreased in both samples during germination, which can be attributed to the drying process that occurs after germination—an essential stage for completing the process. Additionally, extended storage periods contribute to moisture reduction, thereby slowing the growth of germs. The increase in ash content in both samples (red sorghum: 66%, white sorghum: 154%) aligns with the findings of previous studies [22,23]. It has been reported that the increase in ash during germination is due to a decrease in the amount of total soluble dry matter [22]. Discrepancies in results could be attributed to variations in varieties, geographic factors, treatment conditions, and analytical methods. Additionally, [1] found that proteins and carbohydrates in ash act as coenzymes in catalysis and are, therefore, highly consumed.

The amount of starch, crucial for sorghum digestibility, notably decreased during germination (red sorghum: 49%, white sorghum: 58%). The reduced starch content results from the increased activity of amylase and pullulanase enzymes, leading to the breakdown of starch molecules into maltose, maltotriose, and other oligosaccharides. Enzymes such as α-amylase, glucosidase, dextranase (produced in aleurone), and β-amylase (produced in endosperm) are activated during germination and contribute to starch hydrolysis [20,24]. Our study findings are consistent with those reported by [20,22,24].

### 3.2. Mineral Substance Composition with ICP-OES

White- and red-colored sorghum samples were analyzed in terms of macro- (calcium, potassium and magnesium) and microelement (iron and zinc) amounts following a 7-day germination period, and the results are given in Figure 2.

In Figure 2, it is evident that white-colored sorghum samples exhibit a more abundant profile concerning both macro- and micro-element content compared to their red-colored counterparts. Notably, the first day of the germination process resulted in a decrease in the levels of macro- and microelements in both sorghum samples. When considering the germination process as a whole, an overall increase, particularly in the levels of macroelements such as Ca, K, and Mg, was observed as the germination duration extended. 

In the case of white sorghum grains subjected to a 7-day germination process, the calcium content increased from 163.23 mg/L to 584.43 mg/L, and the magnesium content increased from 534.3 mg/L to 643.98 mg/L. However, there was a reduction of 2.3%, 1.6%, and 9.7%, respectively, in the levels of iron, potassium, and zinc compared to the initial raw material. Conversely, red-colored sorghum samples displayed a consistent decrease in elemental content during germination. The calcium content increased from 326.21 mg/L to 292.53 mg/L, the iron content increased from 18.99 mg/L to 15.99 mg/L, the potassium content decreased from 1356.14 mg/L to 975.59 mg/L, and the magnesium content decreased from 500.19 mg/L. The zinc content exhibited a minimal decrease from 13.58 mg/L to 13.46 mg/L in the raw material.

These findings align with other studies in the literature. For instance, a study by [1] on germinated pea flour for 72 h indicated an 11.12% increase in calcium content and an 8.97% decrease in zinc content, which is consistent with our results. In another study, [1] reported that the germination of corn for 24 h led to a 79% increase in iron content and an 80% increase in zinc content. Furthermore, a study by [25] on germinated rice (normal brown, Heukjinju, and Keunnujami) indicated that calcium content increased and potassium content decreased. However, in the case of brown rice, all mineral substance values (calcium, magnesium, potassium, phosphorus) increased with germination. In a study conducted by [26], which investigated the effects of various treatments (soaking, cooking, germination, and fermentation) on sorghum composition, it was reported that the germination process had a diminishing effect on mineral element levels in different sorghum varieties. In the same study, it was noted that the soaking process also contributed to this reduction. Additionally, in a study where two different sorghum cultivars were soaked for 22 h at room temperature and subsequently germinated for 36 and 48 h, it was reported that the mineral content (iron, zinc, and calcium) of both cultivars increased significantly with prolonged germination. This increase was attributed to the loss of solute content in the washing and soaking water [27].

The variation in results among different studies can be attributed to factors such as the phytate content of sorghum, phytase activation, the degree of mineral binding, or the interaction of these factors [28], as indicated in the existing literature data.

### 3.3. Crude Fiber Composition

The Acid Detergent Fiber (ADF), Neutral Detergent Fiber (NDF) and Acid Detergent Lignin (ADL) contents of germinated red- and white-colored sorghum samples are given in Table 3.

The total plant cell wall matrix, comprising major components such as NDF (Neutral Detergent Fiber), cellulose, hemicellulose, and lignin, as delineated in Table 3, plays a crucial role in the structural composition of sorghum grains. The ADF (Acid Detergent Fiber) encompasses lignocellulosic materials composed of lignin and cellulose, while ADL (Acid Detergent Lignin) denotes crude lignin.

The germination process caused an increase in NDF, ADF, and ADL values in both red and white sorghum samples compared to the control samples. This increase, although not consistently linear with respect to the progression of germination time, was found to be approximately 84% and 12%, respectively, for red and white sorghum in terms of NDF values; around 26% and 260% for ADF values; and 3% and 171% for ADL values, respectively, on the seventh day of germination compared to the control samples.

In a study conducted by [29], on germinated lentils, it was reported that germination had a decreasing effect on the NDF amount, primarily due to the reduction in hemicellulose content, while cellulose and lignin amounts (ADF and ADL) increased. Similarly, in another study involving germinated cowpea samples, a decrease in hemicellulose and cellulose content, coupled with an increase in lignin content, was observed [30]. Furthermore, it was found that the crude fiber content increased by 212% in oats germinated for 72 h, as reported by [31]. Another study involving the germination of buckwheat and quinoa for 72 h revealed a 71% increase and a 52% increase, respectively, in total fiber content. These findings are attributed to the disruption of protein–carbohydrate interactions and structural alterations in polysaccharides within the seed cell wall. Additionally, the observed rise in fiber contents during the germination process can be explained by the loss of dry mass resulting from starch hydrolysis through enzymatic action triggered during germination, an increase in lignin content, and the degradation of cellular components such as lipids and proteins [21].

### 3.4. Effect of Germination on Molecular Characterization of Sorghum Samples

FT-IR results showing the change in molecular structures of red- and white-colored sorghum samples during germination are given in Figure 3.

The study data for the determination of the molecular structure of sorghum flour using the FT-IR method revealed the existence of important active regions in the wavenumber range of 1800 to 800 cm^−1^. While FT-IR peaks reveal starch and proteins as the most basic components in sorghum, lipids and phenolic acids are characterized by peaks showing much lesser amounts (Figure 3) [5].

Due to the FT-IR spectra, the peaks were seen at a wavelength of approximately 3400 cm^−1^ for both sorghum samples (between 3420 and 3448 cm^−1^ for white sorghum samples and between 3400 and 3426 cm^−1^ for red sorghum samples). It has been defined as starch and protein molecules that are mainly responsible for the stretching actions of O-H and N-H bonds. The peaks seen at 1652 and 1538 cm^−1^ wavelengths characterize the amide I (C=O) and amide II (N-H, C-N) regions that form the basic structure of the protein molecule [5]. These peaks were detected in the wavelength ranges of 1647–1653 cm^−1^ (amide I) and 1540–1558 cm^−1^ (amide II) for both sorghum cultivars, respectively. The peaks in the wavelength range of 1400–900 cm^−1^ are characterized by another important basic component of sorghum: the starch molecule [5].

FT-IR analysis shows that the germination process causes significant changes in the structures of carbohydrate and protein molecules of white and red sorghum samples. It is seen that this change has gained a more prominent feature depending on the increasing germination time, especially in white sorghum samples. These changes are very important in terms of improving the functional properties of germinated white sorghum samples.

Different intensity peaks seen in the wavelength range of 1400 to 900 cm^−1^ in the FT-IR spectrum were associated with the crystallinity level in sorghum starch due to the change in the amylose/amylopectin ratios forming the starch [32]. With germination, the band ratio in the starch molecule increased at the end of the seventh day compared to the raw material in white sorghum. This is due to the hydrolysis of starch by α- and β-amylase with germination [33]. However, this change was not observed in the red sorghum. This is thought to be due to botanical and genetic differences. Our study results are not similar to the 72 h germination of mung bean and the change in starch [34] but are similar to [33].

Infrared spectrum data show that the germination process causes a significant change in the molecular structure of white and red sorghum grains. Different germination times caused different rates of change in both sorghums. The germination process in white sorghum samples for 24 h caused the most change in the molecular structure of the samples compared to the control sample, while this period was determined as 4 days for red sorghum samples [35]. Groups of observers of germinated HomChaiya rice reported that it peaked at day 7 in the sequence of enzymatic hydrolysis of the outbreak, which occurred on day 5, which is in line with our study. 

### 3.5. SEM Images of Sorghum Samples during Germination

SEM images of white and red sorghum samples are given in Figure 4 to better reflect the change in their molecular structures during the 7-day germination period. 

The data show that there is a very strong destruction of starch granules and protein structures compared to control samples due to increased germination time. While it is possible to see individual starch granules surrounded by proteins, especially in the control samples, at the end of the 7-day germination period, the protein structures separate from the starch granules and lose their visibility. Starch granules, on the other hand, are eroded in the regions where the proteins are restricted. The study data are in agreement with the FT-IR results showing a decrease in the amount of protein and a change in the amount of starch. The effects of the germination process on red and white sorghum grains are in agreement with similar studies [36]. Enzymatic hydrolysis occurs during grain germination, and starch is released from the interconnection and entanglement of proteins and fibers, changing the structure of flour. Protein and fiber are hydrolyzed into amino acids and glucose [36]. The enzyme activity occurs during germination. Initially, starch hydrolysis starts after the enzyme adheres to the starch surface. The channels that allow the enzyme to spread into the granule center widen in the second stage as hydrolysis progresses and the number of tiny holes on the granule surface rises. The catalytic enzyme action degrades and modifies the granule surface in the final stage. Moreover, as a result of the change in molecular structure brought about by the rise in granule porosity, the crystallinity of the granules decreases [22]. 

In a previous study, it was reported that the polyhedral starch that adhered to the protein matrix in barley grain with germination was separated from the first day. This is due to the increase in enzymatic activity with germination. It was also reported in the study that the maximum activity of α-amylase occurs between the third and fifth days, while the protease can start from the fifth day of germination. Depending on the germination conditions and seed viability, the varying enzymatic activity has an impact on the various changes in the morphology of the grain throughout the germination period. According to several studies, the starch granules in cereals dissolve during germination, leaving them with rough surfaces and pinholes. The eroding of the granule surface or the digestion of the channels from the chosen locations on the surface towards the center are thought to be the causes of partial hydrolysis of starch. The uneven modification of the surface might be the result of variation in the susceptibility to hydrolytic enzyme activity during germination. The enzyme spreads to the solid surface and is adsorbed through the sites; finally, the catalytic reaction is produced. Adsorption can be affected by granule size, surface characteristics, and minor constituents, such as proteins and lipids on the surface [22,33]. The effect of germination on morphological traits is compatible with [22].

### 3.6. Color Values of Sorghum Samples 

The grain color is related to pigments in the testa and pericarp [37]. The phenolic compounds (tannins and anthocyanins) or starch granules are the color pigments in the sorghum pericarp [38]. The color changes during the germination in red and white sorghum are presented in Table 4. The *L** value indicates the lightness of the flour and ranges between 0 (black) and 100 (white). *L** values were in the range between 73.50 and 1.41 (red) and 80.27 and 2.25 (white) in ungerminated sorghum flour. The lightness values of red and white sorghum were decreased significantly with the germination process (*p* < 0.05) and were measured similar in both grains after 7 days of germination (Table 4). [39] stated that the decreasing lightness value associated with the increasing protein content during germination. *L** values of raw wheat (85.29) and sorghum (86.58) flour were comparable with the lightness of ungerminated white sorghum [40]. Flour color generally affects the color of finished products, and high *L** values are more desirable for many baking goods [41]. The *L** value (72.04 ± 1.38) of a cookie made from wheat flour was found higher compared to a cookie made from germinated (50.01 ± 1.99) and ungerminated (59.13 ± 2.45) minor millet flour [39]. Positive *a** and *b** values indicate redness and yellowness in the sample. a* values were 2.30 and 5.10 in raw white and red sorghum, respectively. The *a** values in white and red sorghum grains increased during the germination process, while the *b** values did not change significantly (*p* > 0.05). 

### 3.7. Total Phenolic Content

The effects of germination on the red- and white-pigmented pericarp sorghum grains exhibited significant variation in TPC at *p* < 0.05 (Figure 5). 

The phenolic content in red sorghum varied from 89.76 ± 8.31 mg GAE/100 g to 379.5 ± 6.92 mg GAE/100 g during the germination. After 72 h of germination, TPC was increased three times in red sorghum when compared to the ungerminated control sample (136.64 ± 3.76 mg GAE/100 g). Similarly, the phenolic content of white sorghum increased significantly (*p* < 0.05) from 52.84 ± 3.31 mg GAE/100 g (C_W_) to 151.76 mg GAE/100g (G_W7_). The red- and white-pigmented pericarp sorghum grains exhibited a significant difference in TPC, while similar increase ratios were observed for phenolic content after germination. [42] demonstrated that the black pericarp sorghum variety had the highest TPC at 11.50 ± 1.81 mg GAE /g, followed by the brown pericarp sorghum at 3.58 ± 1.63 mg GAE/g. However, the red- and white-pigmented pericarp varieties did not exhibit a significant difference in TPC. The results are comparable with the findings of [43], who reported a 39.74% increase in the TPC after 48 h of germination in sorghum. Many studies showed that germination processes clearly increase TPC in quinoa, amaranth, buckwheat, and millet [21,44,45]. Phenolic compounds can be found in free or bound forms in cereals. Free phenolic compounds are located in the pericarp and can usually be extracted with an organic solvent. It is thought that the total amount of phenolics produced by the germination process may be due to the oxidation of endogenous phenolics by the activity of enzymes such as peroxidase and polyphenol oxidase [46]. 

### 3.8. Radical Scavenging Activity

The DPPH free-radical-scavenging activity of red- and white-pigmented pericarp sorghum is presented in Figure 5. 

The red sorghum grain had significantly higher antioxidant activity in comparison to the white sorghum grains at *p* < 0.05. The germination process was significantly effective in both red and white grains and reached its highest level (184.94 mg AAE/100g and 55.46 mg AAE/100g, respectively) after 7 days of germination. Previous studies have also indicated that germination has an enhancing impact on the DPPH scavenging activity of sorghum grains [43,47]. The DPPH scavenging activity of germinated wheat, barley, and millet grain was increased significantly from 1.37–1.64 g AAE/g to 3.19–3.76 g AAE/g after 72 h of germination [48]. The antioxidant activity (DPPH) of germinated amaranth, quinoa, and buckwheat grains also was significantly higher (35–178%) than that of ungerminated grains [21]. The antioxidant potential of cereal grains is mostly derived from phenolic compounds, which are also crucial in the prevention and management of degenerative illnesses. This study demonstrates that germination offers a novel strategy for advancing the development of sorghum seed as a useful food for human consumption.

## 4. Conclusions

Sorghum, due to its rich composition of bioactive compounds, low digestibility, high antioxidant capacity, and its resilience to drought conditions among cereal crops, is considered a highly important food resource for future human nutrition. This study investigates the impact of germination, specifically over a period of 7 days, on red and white sorghum grains, focusing on pivotal parameters associated with the germination process such as nutritional fiber content, protein and starch characteristics, and bioactive compounds. 

During germination, there was a notable increase in protein and lipid content, accompanied by a decrease in starch content. White sorghum grains exhibited heightened levels of calcium and magnesium but experienced reductions in iron, potassium, and zinc. Conversely, red sorghum grains consistently displayed a decline in mineral content throughout germination. Prolonged germination durations correlated with elevated values of NDF, ADF, and ADL in both red and white sorghum samples when compared to control groups. This escalation in dietary fiber content holds significance for the functional attributes of sorghum grains with regards to health considerations. 

Germination led to the separation of starch granules from their protein coatings and increased the damage to starch granules. Starch granules with altered molecular structures due to germination are transformed into healthier forms for nutrition. With a 7-day germination period, total phenolic content and antioxidant activity reached their maximum levels in both red and white sorghum, and the difference compared to the control group was statistically significant. After 72 h of germination, the TPC in germinated seeds exhibited a threefold increase compared to ungerminated seeds. 

This study is highly significant for increasing the potential use of red and white sorghum varieties, which are currently only used as animal feed in most countries such as Turkey, in human nutrition by subjecting them to the germination process and determining the effects of germination at different durations on the functional composition of sorghum. During the 7-day germination period, an increase was observed in all parameters analyzed within the scope of the study, and the germination process positively affected the functional properties that determine the positive effects of both white and red sorghum samples on health. From a future perspective, digestibility experiments are needed to determine the optimum germination time more precisely. In particular, investigating the digestibility of sorghum proteins during germination is highly important.

## Figures and Tables

**Figure 1 foods-13-00662-f001:**
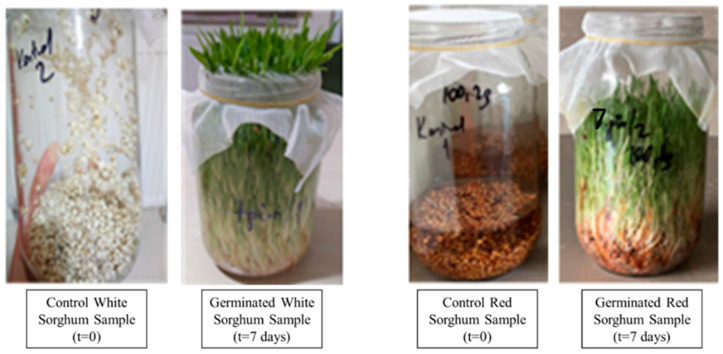
Images of germinated red and white sorghum grains.

**Figure 2 foods-13-00662-f002:**
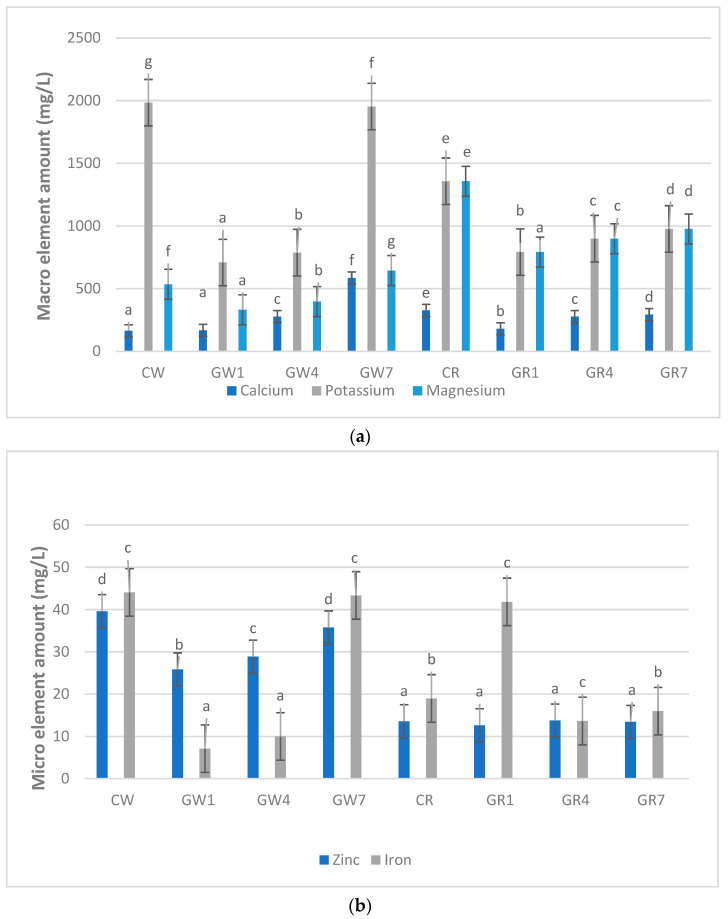
Effect of germination process on macro- (**a**) and micro- (**b**) element distribution of white and red sorghum samples. Data (g/100 g DM) are expressed as mean values ± standard deviation. Means marked with the same letter are not statistically different from each other (*p* < 0.05) C_w_: Control White Sorghum, G_W1_: Germination Sorghum White 1 days, G_W4_: Germination Sorghum White 4 days, G_W7_: Germination Sorghum White 7 days, C_R_: Control Red Sorghum, G_R1_: Germination Sorghum Red 1 day, G_R4_: Germination Sorghum Red 4 days, G_R7_: Germination Sorghum Red 7 days.

**Figure 3 foods-13-00662-f003:**
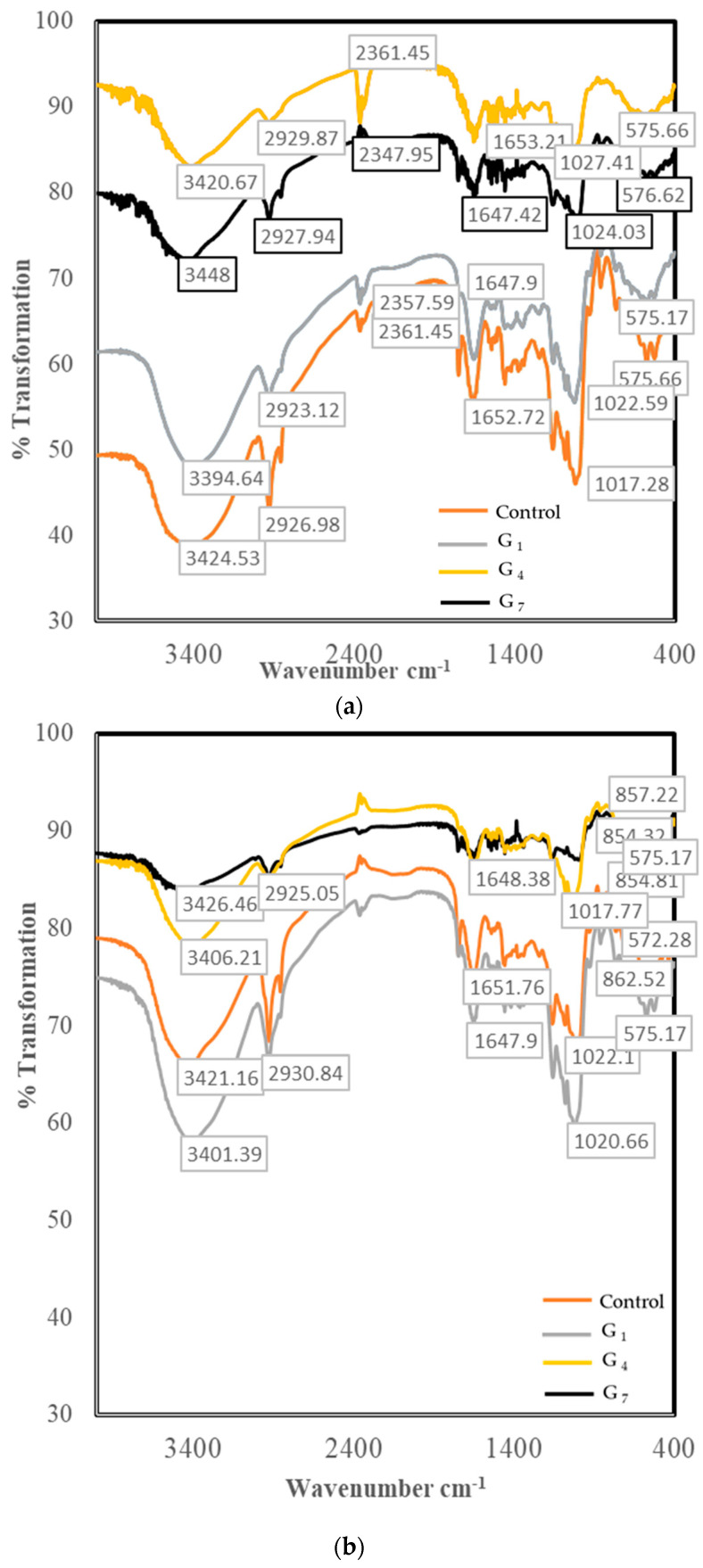
Effect of germination process on FT-IR of white (**a**) and red (**b**) sorghum samples.

**Figure 4 foods-13-00662-f004:**
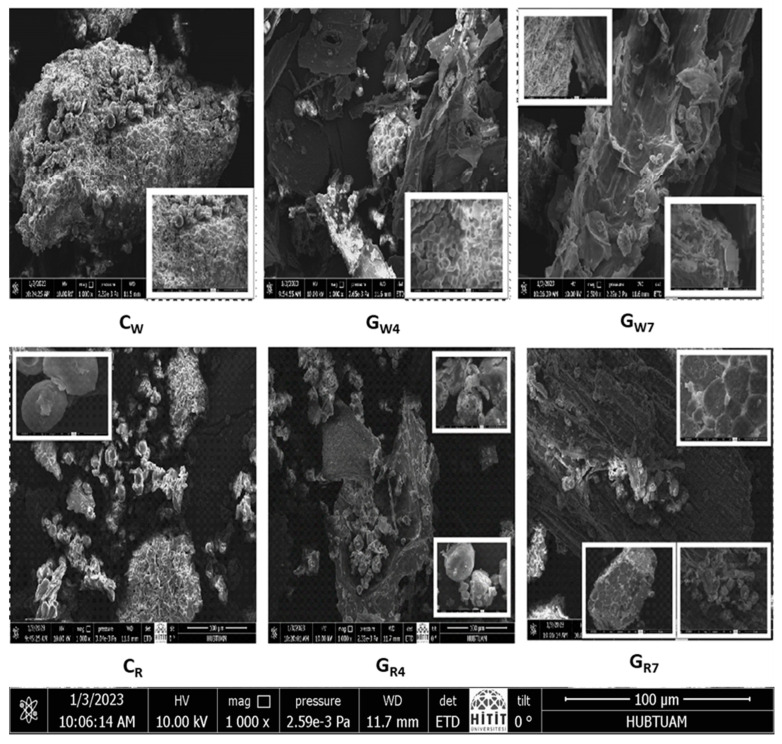
SEM images of white and red sorghum samples during germination process. C_w_: Control White Sorghum, G_W4_: Germination Sorghum White 4th day, G_W7_: Germination Sorghum White 7th day, C_R_: Control Red Sorghum, G_R4_: Germination Sorghum Red 4th day, G_R7_: Germination Sorghum Red 7th day.

**Figure 5 foods-13-00662-f005:**
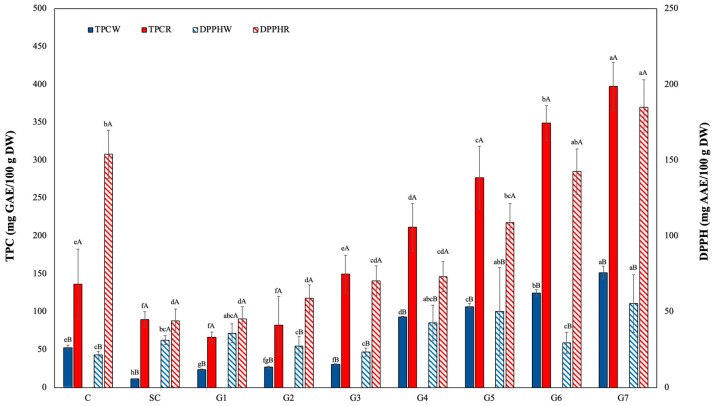
Effects of germination on total phenolic and antioxidant activity of sorghum samples. a−f: Lowercase letters indicate the effect of germination time on the TPC and DPPH value. Means followed by the same letter do not differ significantly at *p* = 0.05 according to Tukey multiple ranges; A−B: capital letters indicate the effect of sorghum variety at the same germination time. Means followed by the same letter do not differ significantly at *p* = 0.05, according to the t-test.

**Table 1 foods-13-00662-t001:** ICP-OES operating conditions.

Operating Conditions	
Power	1150 W
Auxiliary Gas Flow	0.5 L/min
Coolant gas	12 L/min
Nebulizer gas	0.70 L/min

**Table 2 foods-13-00662-t002:** The proximate chemical composition of red and white sorghum grains.

Sample	Moisture	Ash	Protein	Lipid	Starch
**C_W_**	11.31 ± 0.06 ^m^	2.05 ± 0.05 ^b^	12.58 ± 0.59 ^g^	3.49 ± 0.05 ^b^	53.71 ± 3.42 ^j^
**SC_W_**	2.95 ± 0.20 ^cd^	2.43 ± 0.18 ^c^	13.49 ± 0.12 ^h^	4.52 ± 0.42 ^ef^	46.76 ± 0.23 ^h^
**G_W1_**	2.34 ± 0.11 ^b^	2.88 ± 0.05 ^de^	12.14 ± 0.16 ^fg^	4.49 ± 0.64 ^ef^	44.9 ± 0.43 ^g^
**G_W2_**	3.49 ± 0.34 ^ef^	3.15 ± 0.19 ^ef^	11.88 ± 0.67 ^f^	4.85 ± 0.17 ^fg^	41.41 ± 1.49 ^f^
**G_W3_**	4.24 ± 0.31 ^g^	3.23 ± 0.12 ^f^	12.20 ± 0.32 ^fg^	5.28 ± 0.15 ^h^	40.59 ± 1.59 ^f^
**G_W4_**	6.37 ± 0.49 ^i^	3.60 ± 0.14 ^g^	10.82 ± 0.65 ^e^	5.41 ± 0.21 ^hi^	35.19 ± 0.38 ^d^
**G_W5_**	8.04 ± 0.33 ^j^	4.43 ± 0.09 ^i^	11.73 ± 0.61 ^f^	5.79 ± 0.08 ^ij^	27.80 ± 1.84 ^b^
**G_W6_**	8.04 ± 0.33 ^j^	4.75 ± 0.1 ^j^	14.35 ± 0.04 ^i^	6.08 ± 0.28 ^j^	26.65 ± 0.62 ^b^
**G_W7_**	8.85 ± 0.54 ^k^	5.20 ± 0.08 ^k^	15.02 ± 0.05 ^j^	6.12 ± 0.25 ^j^	22.59 ± 0.66 ^a^
**C_R_**	11.35 ± 0.057 ^m^	1.25 ± 0.06 ^a^	9.91 ± 0.57 ^abcd^	2.26 ± 0.14 ^a^	61.87 ± 0.04 ^k^
**SC_R_**	2.44 ± 0.31 ^m^	3.10 ± 0.69 ^ef^	10.13 ± 0.52 ^bcd^	3.74 ± 0.56bc	49.06 ± 0.46 ^i^
**G_R1_**	1.90 ± 0.063 ^a^	2.78 ± 0.09 ^d^	10.30 ± 0.07 ^cde^	3.42 ± 0.06 ^b^	43.44 ± 0.57 ^g^
**G_R2_**	2.62 ± 0.075 ^bc^	2.70 ± 0.18 ^cd^	9.45 ± 0.28 ^a^	3.64 ± 0.23 ^bc^	45.04 ± 1.49 ^g^
**G_R3_**	3.11 ± 0.13 ^de^	3.00 ± 0.08 ^def^	9.87 ± 0.06 ^abcd^	4.04 ± 0.31 ^cd^	40.97 ± 0.32 ^f^
**G_R4_**	3.83 ± 0.11 ^fg^	3.23 ± 0.05 ^f^	9.74 ± 0.05 ^abc^	4.3 ± 0.13 ^de^	38.8 ± 0.30 ^e^
**G_R5_**	4.19 ± 0.29 ^g^	3.28 ± 0.05 ^f^	9.61 ± 0.22 ^ab^	4.61 ± 0.07 ^ef^	37.79 ± 0.24 ^e^
**G_R6_**	5.54 ± 0.12 ^h^	3.30 ± 0.00 ^f^	10.39 ± 0.24 ^de^	5.14 ± 0.12 ^gh^	35.42 ± 0.50 ^d^
**G_R7_**	5.97 ± 0.56 ^i^	4.13 ± 0.09 ^h^	12.28 ± 0.04 ^fg^	5.33 ± 0.22 ^h^	31.59 ± 0.35 ^c^

Data (g/100 g DM) are expressed as mean values ± standard deviation. Means marked with the same letter are not statistically different from each other (*p* < 0.05) C_R_: Control Red Sorghum, SC_R_: Soak Control Red Sorghum, G_R1_: Germination Sorghum Red 1 day, G_R7_: Germination Sorghum Red 7 days; C_w_: Control White Sorghum, SC_W:_ Soak Control White Sorghum, G_W1_: Germination Sorghum White 1 days, G_W7_: Germination Sorghum White 7 days.

**Table 3 foods-13-00662-t003:** The effect of germination process on dietary fiber composition of sorghum samples.

Sample	Neutral Detergent Fiber (NDF, %)	Acid Detergent Fiber (ADF, %)	Acid Detergent Lignin (ADL, %)
**C_W_**	25.08 ± 1.23 ^d^	2.73 ± 0.40 ^bc^	0.43 ± 0.07 ^abcd^
**SC_W_**	25.51 ± 0.69 ^d^	0.97 ± 0.58 ^ab^	0.26 ± 0.10 ^a^
**G_W1_**	15.98 ± 0.95 ^bc^	0.74 ± 0.54 ^a^	0.40 ± 0.08 ^abcd^
**G_W2_**	11.10 ± 0.91 ^ab^	1.20 ± 0.32 ^ab^	0.33 ± 0.13 ^ab^
**G_W3_**	11.30 ± 2.07 ^ab^	0.98 ± 0.33 ^ab^	0.35 ± 0.12 ^abc^
**G_W4_**	24.02 ± 5.28 ^d^	3.09 ± 0.70 ^c^	0.67 ± 0.01 ^abcdef^
**G_W5_**	30.69 ± 6.57 ^e^	5.82 ± 0.50 ^d^	1.02 ± 0.31 ^ef^
**G_W6_**	28.62 ± 0.35 ^de^	6.06 ± 0.34 ^de^	0.93 ± 0.21 ^ef^
**G_W7_**	29.05 ± 0.61 ^de^	8.81 ± 1.93 ^fg^	1.33 ± 0.43 ^f^
**C_R_**	9.30 ± 0.14 ^a^	9.30 ± 0.14 ^fg^	1.06 ± 0.05 ^ef^
**SC_R_**	7.72 ± 1.09 ^a^	7.72 ± 1.08 ^efg^	0.94 ± 0.00 ^ef^
**G_R1_**	7.84 ± 1.02 ^a^	7.83 ± 1.02 ^fg^	0.95 ± 0.08 ^ef^
**G_R2_**	7.17 ± 0.19 ^a^	7.16 ± 0.19 ^def^	0.61 ± 0.41 ^abcde^
**G_R3_**	11.46 ± 0.29 ^ab^	11.46 ± 0.29 ^h^	0.80 ± 0.05 ^cdef^
**G_R4_**	12.04 ± 1.56 ^abc^	12.04 ± 1.55 ^h^	0.75 ± 0.28 ^bcdef^
**G_R5_**	11.41 ± 0.60 ^ab^	11.40 ± 0.60 ^g^	0.95 ± 0.09 ^ef^
**G_R6_**	14.95 ± 0.05 ^bc^	14.95 ± 0.05 ^h^	0.85 ± 0.01 ^def^
**G_R7_**	16.91 ± 0.36 ^c^	16.91 ± 0.36 ^i^	1.13 ± 0.01 ^ef^

Data (g/100 g DM) are expressed as mean values ± standard deviation. Means marked with the same letter are not statistically different from each other (*p* < 0.05). C_R_: Control Red Sorghum, SC_R_: Soak Control Red Sorghum, G_R1_: Germination Sorghum Red 1 day, G_R7_: Germination Sorghum Red 7 days; C_w_: Control White Sorghum, SC_W:_ Soak Control White Sorghum, G_W1_: Germination Sorghum White 1 days, G_W7_: Germination Sorghum White 7 days.

**Table 4 foods-13-00662-t004:** Effects of germination on CIE color parameters of red and white sorghum samples.

	*L**	*a**	*b**	*C**	*h*º
**C_W_**	80.27 ± 2.25 ^ab^	2.30 ± 0.46 ^c^	13.74 ± 0.74 ^a^	13.93 ± 0.81 ^a^	80.57 ± 1.36 ^a^
**SC_W_**	80.58 ± 1.58 ^ab^	2.03 ± 0.30 ^c^	14.09 ± 0.28 ^a^	14.24 ± 0.30 ^a^	81.81 ± 1.12 ^a^
**G_W1_**	82.28 ± 1.22 ^a^	1.78 ± 0.17 ^c^	14.15 ± 0.60 ^a^	14.26 ± 0.62 ^a^	82.83 ± 0.37 ^a^
**G_W2_**	82.25 ± 1.09 ^a^	1.94 ± 0.14 ^c^	13.63 ± 0.41 ^a^	13.76 ± 0.42 ^a^	81.93 ± 0.43 ^a^
**G_W3_**	80.61 ± 0.70 ^ab^	2.28 ± 0.14 ^c^	13.82 ± 0.19 ^a^	14.01 ± 0.20 ^a^	80.65 ± 0.50 ^a^
**G_W4_**	78.00 ± 1.34 ^bc^	2.40 ± 0.22 ^c^	13.85 ± 0.84 ^a^	14.06 ± 0.81 ^a^	80.11 ± 1.31 ^a^
**G_W5_**	75.44 ± 0.94 ^cd^	3.39 ± 0.45 ^b^	13.03 ± 0.22 ^a^	13.47 ± 0.28 ^a^	75.44 ± 1.76 ^b^
**G_W6_**	73.68 ± 0.33 ^d^	3.76 ± 0.18 ^ab^	12.97 ± 0.72 ^a^	13.50 ± 0.66 ^a^	73.80 ± 1.43 ^bc^
**G_W7_**	69.55 ± 1.36 ^e^	4.47 ± 0.71 ^a^	13.40 ± 0.92 ^a^	14.15 ± 0.77 ^a^	71.47 ± 3.50 ^c^
**C_R_**	73.50 ± 1.41 ^abc^	5.10 ± 0.54 ^b^	11.47 ± 0.68 ^b^	12.55 ± 0.85 ^b^	66.07 ± 1.02 ^ab^
**SC_R_**	71.35 ± 1.07 ^bcd^	5.24 ± 0.20 ^b^	12.36 ± 0.70 ^ab^	13.42 ± 0.71 ^ab^	67.02 ± 0.53 ^a^
**G_R1_**	70.34 ± 0.93 ^cd^	5.74 ± 0.21 ^b^	13.36 ± 0.30 ^a^	14.54 ± 0.32 ^a^	66.73 ± 0.63 ^a^
**G_R2_**	72.97 ± 1.14 ^abc^	5.55 ± 0.19 ^b^	12.23 ± 0.59 ^ab^	13.43 ± 0.58 ^ab^	65.57 ± 0.92 ^ab^
**G_R3_**	73.59 ± 1.05 ^ab^	5.58 ± 0.45 ^b^	12.52 ± 0.40 ^ab^	13.71 ± 0.48 ^ab^	66.01 ± 1.57 ^ab^
**G_R4_**	74.11 ± 0.68 ^a^	5.47 ± 0.14 ^b^	11.91 ± 0.54 ^b^	13.10 ± 0.53 ^b^	65.33 ± 0.81 ^ab^
**G_R5_**	73.73 ± 1.52 ^ab^	5.97 ± 0.58 ^b^	11.69 ± 0.43 ^b^	13.13 ± 0.65 ^b^	63.02 ± 1.46 ^bc^
**G_R6_**	70.66 ± 1.39 ^cd^	6.95 ± 0.20 ^ab^	11.95 ± 0.28 ^b^	13.82 ± 0.21 ^ab^	59.82 ± 1.14 ^cd^
**G_R7_**	69.54 ± 0.63 ^d^	7.22 ± 0.70 ^a^	12.03 ± 0.52 ^b^	14.04 ± 0.39 ^ab^	59.03 ± 3.20 ^d^

Data are presented as the “mean ± standard deviation” of three independent experiments (n = 3). a−d: Lowercase letters indicate the effect of germination time on the CIE color value. Means followed by the same letter do not differ significantly at *p* = 0.05 according to Tukey multiple ranges.

## Data Availability

The original contributions presented in the study are included in the article, further inquiries can be directed to the corresponding author.

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
