# Peer review of "Germination: A Powerful Way to Improve the Nutritional, Functional, and Molecular Properties of White- and Red-Colored Sorghum Grains"

_foods, 2024, doi:10.3390/foods13050662_

Round 1
Reviewer 1 Report
Comments and Suggestions for Authors
1. In the introduction, it is necessary to further expand the review on the effects of germination treatment on plants.
2. Line 92, L should not be italicized.
3. Figure 1 is not essential, and it is recommended to delete it.
4. Line 117, please provide detailed explanations of the measurement methods, particularly the sample handling methods. Additionally, I would like to know the number of seedlings collected from 500g samples at each germination stage, as well as the total number of germination treatments conducted by the authors for measurement purposes.
5. Line 109, why were there only two repetitions for the germination treatment?
6. Line 190, please provide photos of the seedlings at each germination period (for review only).
7. The data in Figure 3 needs to be annotated with standard deviation and subjected to significance testing. Additionally, the results description and discussion for Figure 3 need to be rewritten.
8. What part of the seedling is captured in the SEM images? Figure 4 needs to be enhanced in terms of clarity, and the relevant parameters and scale should be indicated.
9. The authors need to provide a detailed description of how the samples were handled during the measurement of various indicators. Were the whole seedlings or specific tissues measured for indicators such as color and TPC?
10. It is recommended to verify the multiple comparison results in Figure 6.
11. Line 449 and line 23, "There was a 3 times increase in TPC after 72 h of germination". Is it a comparison between G3 and C?
12. How was the longest germination time (7d) determined? In what specific aspects can the seedlings at 7 days of germination be applied? How are they prepared for application?
13. The author needs to carefully adjust the references, such as italicizing the species names in the references. Where can I find the references mentioned in line 454 and line 85?
14. Table S1 is not essential, and the significance results between varieties need to be annotated in Figure 6.
Author Response
Dear Reviewer 1,
I hope this message finds you well. I wanted to extend my sincerest gratitude for taking the time to review my manuscript. Your thoughtful insights and constructive feedback have been invaluable in refining the content and strengthening the overall quality of the manuscript.
Your thorough review demonstrated a deep understanding of the subject matter, and I truly appreciate the time and effort you invested in providing detailed comments and suggestions. Your expertise and insights have undoubtedly enhanced the clarity, coherence, and rigor of the manuscript.
I am deeply grateful for the opportunity to benefit from your expertise and insights.
Warm regards,

Reviewer 2 Report
Comments and Suggestions for Authors
The germination time is very short, and the data were collected for only seven days. I believe extending the observation period to 30 days would be beneficial, as accurate analysis may be challenging within the first seven days. In my opinion, this represents a significant flaw in the experiment. Seven days is also beneficial in case if seed grain analyze before germination. It would be more advantageous to explore the nutritional composition of the seed grain both before germination. I recommend that the authors also analyze red and white grains before germination, as this would enhance the study's robustness.
There are too many grammatical mistakes found in the paper, review thoroughly
Comments on the Quality of English Language
Need minor English revision
Author Response
Dear Reviewer 2,
I hope this message finds you well. I wanted to extend my sincerest gratitude for taking the time to review my manuscript. Your thoughtful insights and constructive feedback have been invaluable in refining the content and strengthening the overall quality of the manuscript.
Your thorough review demonstrated a deep understanding of the subject matter, and I truly appreciate the time and effort you invested in providing detailed comments and suggestions. Your expertise and insights have undoubtedly enhanced the clarity, coherence, and rigor of the manuscript.
I am deeply grateful for the opportunity to benefit from your expertise and insights.
Warm regards,

Reviewer 3 Report
Comments and Suggestions for Authors
Article submitted for review: „ Germination: A powerful way to improve the nutritional, func-2 tional, and molecular properties of red and white-colored sorghum grains” ontains important information on the change in the chemical composition of sorghum grain of two varieties after germination. The purpose of the research is well justified in the introduction. The methodology is clearly written. Results well discussed and explained. The conclusions are justified by the content of the work. Basically, this manuscript is well written and results were discussed properly. However, I have some questions about this work, as follow:
L49: “…with essential amino acids being abundant” What are the essential amino acids in sorghum grain?
L94: What was the air humidity?
L101: “Thanks to soaking, microbial load was decreased..” This is not obvious. Please explain.
L192: Table 2. What are the %? g / 100 g grain? g / 100 g dm? Explain!
L225: What could be the reason for the increase in fat content?
L227-228: What drying process occurs during germination?
L297: Table 3. % -?
L337: Fig 4. The legend is missing. Which color means what?
The figures show the data in the following order: white sorghum, red sorghum. In the tables, the order is reversed, which makes reading very difficult. The title of the work is red and white sorghum grains. This can be standardized.
Comments on the Quality of English Language
-
Author Response
Dear Reviewer 3,
I hope this message finds you well. I wanted to extend my sincerest gratitude for taking the time to review my manuscript. Your thoughtful insights and constructive feedback have been invaluable in refining the content and strengthening the overall quality of the manuscript.
Your thorough review demonstrated a deep understanding of the subject matter, and I truly appreciate the time and effort you invested in providing detailed comments and suggestions. Your expertise and insights have undoubtedly enhanced the clarity, coherence, and rigor of the manuscript.
I am deeply grateful for the opportunity to benefit from your expertise and insights.
Warm regards,

Round 2
Reviewer 1 Report
Comments and Suggestions for Authors
The paper has been carefully revised and can be accepted for publication.
Author Response
Dear Reviewer
Thank you for your invaluable support and collaboration. I am deeply grateful for the opportunity to benefit from your expertise and insights.
Warm regards,
Dr Secil UZEL
Reviewer 2 Report
Comments and Suggestions for Authors
Agreed with author response
Author Response

(The authors gave the same response as above.)
